# Effectiveness of Lifestyle Interventions for Prevention of Harmful Weight Gain among Adolescents from Ethnic Minorities: A Systematic Review

**DOI:** 10.3390/ijerph17176059

**Published:** 2020-08-20

**Authors:** Nematullah Hayba, Samiha Elkheir, Jessica Hu, Margaret Allman-Farinelli

**Affiliations:** Discipline of Nutrition and Dietetics, School of Life and Environmental Science, Charles Perkins Centre, University of Sydney, Sydney 2006, Australia; samihaelkheir@gmail.com (S.E.); jehu0465@gmail.com (J.H.); margaret.allman-farinelli@sydney.edu.au (M.A.-F.)

**Keywords:** overweight, obesity, prevention intervention, lifestyle, adolescents, ethnic minorities, racial minorities

## Abstract

The escalating obesity among adolescents is of major concern, especially among those from an ethnic minority background. The adolescent period offers a key opportunity for the implementation of positive lifestyle behaviours as children transition to adulthood. The objective of this review was to examine the effectiveness of lifestyle interventions for adolescents and their impact in ethnic and racial minorities for the prevention of overweight and obesity. Seven electronic databases were searched from 2005 until March 2019 for randomized controlled trials of lifestyle programs conducted in this population. The main outcome was change in Body Mass Index (BMI) z-score (kg/m^2^) or change in BMI and secondary outcomes were changes in physical activity and diet. Thirty studies met the inclusion criteria. Seven studies reported and/or conducted subgroup analysis to determine if ethnic/racial group affected weight change. None demonstrated an overall decrease in BMI z-score. However, six of the seven demonstrated changes in secondary measures such as fruit and vegetable intake and screen time. Results did not differ by ethnic/racial group for primary and secondary outcomes. Overweight and obesity prevention among adolescents from ethnic minorities is an area that needs further research. There is a lack of interventions that include analyses of effectiveness in ethnic minorities.

## 1. Introduction

The World Health Organisation recognizes the role of primary prevention of obesity in childhood and adolescence is critical to halt the early onset of chronic diseases in adulthood [1]. Rates are alarmingly high in many countries, with data showing nearly 1 in 5 children and adolescents aged 6–19 years suffer from obesity in the United States [2], and almost a third aged 5–17 years are overweight or obese in Australia [3]. Not only is the quality of life compromised in adulthood, but increasing healthcare costs of resultant non-communicable diseases place a greater financial burden on our healthcare systems [4,5].

Adolescence is a complex life stage as adequate nutrition is vital for growth and development; but poor diet quality may have detrimental effects leading to obesity and other risk factors for chronic disease [6]. The transition from childhood to adulthood involves the emergence of a sense of autonomy that might precipitate resistance to perceived authoritarian healthy lifestyle programs [7]. Moreover, adolescents present with the unhealthiest diets of any age group [8,9] and many do not meet national guidelines for physical activity [10,11]. While there have been numerous programs developed and tested, mostly in schools, they have not always been designed for inclusion of ethnic minorities [6]. Furthermore, the burden of obesity is not equally shared by all sectors of society, with ethnic minority groups most affected [6,12]. This has been shown to be an international phenomenon, with increased obesity rates concentrated in ethnic minorities in several developed countries. For example, in the USA, Hispanics (25.8%) and non-Hispanic blacks (22.0%) are reported to have higher obesity prevalence than non-Hispanic whites (14.1%) among children and adolescents aged 2 to 19 years of age [13]. In the UK, children from black and Asian backgrounds had higher proportions of overweight and obesity, and were three times more likely to present with obesogenic lifestyles than their white counterparts [14]. Adolescents in Australia from the Middle East, North Africa and Oceania regions are exhibiting greater prevalence of overweight and obesity than those from English speaking countries [12,15].

Moreover, U.S. minority groups are most effected by elevated rates of overweight and obesity, as they may also transition away from healthy weight at a younger age when compared to the White population. A multi-ethnic study drawing on data from the U.S. National Health and Nutrition Examination Survey spanning early childhood to late adulthood demonstrated obesity disparities to be evident by two years of age especially for African American females and Mexican American males and females. Disparities in rates of obesity and early transitions to obesity are concerning, given the research demonstrating the difficulties in returning to a normal body weight once an individual has overweight or obesity and inequalities become exacerbated if left unaddressed. This necessitates the additional research needed to prevent obesity during this critical age epoch [16].

To our knowledge, no systematic review focusing on the effectiveness of lifestyle interventions for the prevention of harmful weight gain leading to overweight and obesity in ethnic and racial at-risk populations has been published. Therefore, the aim of this review is to (i) systematically examine the effectiveness of randomized controlled trials (RCTs) of lifestyle interventions for the prevention of overweight and obesity in adolescents and determine how many targeted or included racial and ethnic minorities; and (ii) review the impact of such interventions for adolescents from racial and ethnic minority groups. For the purpose of this review, U.S. definitions of racial and ethnic minorities were used, with American Indians/Alaskan Natives and non-Hispanic Blacks to be considered a racial minority and Hispanics to be an ethnic minority. Race is defined on the basis of physical differences that groups or cultures consider, whereas ethnicity encompasses shared cultural norms such as beliefs, practices, language and ancestry [17].

## 2. Materials and Methods

This systematic review adheres to the Preferred Reporting Items for Systematic Reviews and Meta-Analyses (PRISMA) statement [18] and is registered with PROSPERO (registration number: CRD42018092825).

### 2.1. Identification of Studies

Seven databases were used for a systematic literature search; Medline, PsycINFO, Eric, CINAHL, Web of Science, Embase and Cochrane Central Register of Controlled Trials. The databases were searched from January 2005 until March 2019. Articles published prior to the year 2005 were excluded due to the changes in technology, lifestyle, diet and physical activity that have occurred since then. The reference lists of articles located from the search were also hand searched for any additional missed papers. The search terms were “adolescents”, “lifestyle”, “prevention”, “intervention”, “weight loss”, “weight changes”, “diet” and “physical activity”, and synonyms of these were included. The terms were broad and combined and truncated to encompass all studies that may fit the population, intervention, study design and outcome criteria, therefore lowering the chance of exclusion of any studies that may be eligible. A complete search strategy used in the electronic database Medline is shown in Appendix A.

### 2.2. Eligibility Criteria

Criteria for inclusion were:

Population: study population of adolescents defined as from 13 to 18 years of age. Healthy adolescents classified in the healthy or overweight range, free of any chronic or acute disease which may affect measured outcomes. The authors were contacted for information on ethnic variations in the study population.

Intervention: Interventions with an aim of preventing harmful weight gain and incident overweight and obesity among adolescents by improving lifestyle factors including nutrition and physical activity.

Comparator: Non-exposed group or alternate treatment.

Outcomes: The primary outcome was reported change in BMI z-score (kg/m^−2^) or change in weight, and secondary outcomes in obesity-related measures such as waist circumference and diet and physical activity levels.

Study design: randomized controlled trials (RCTs) including feasibility and pilot trials that were published after the year 2005 and in English. A comprehensive review of weight gain prevention interventions in adolescents was published in 2005 [19]. We selected 2005 as our start date because of both the previous review and acknowledging the significant advancements in technology and health promotion that have occurred since then, with nearly 100% of public schools in the U.S. having access to Internet in 2005 [20]. RCTs were selected to ensure that this review summarises and highlights evidence of higher quality.

Criteria for exclusion were:

Those interventions in children outside the specified age range and those that targeted obese participants (BMI z-score > +2 SD). Studies were also excluded if (i) the interventions aimed to maintain previous weight loss; (ii) the interventions were in children with chronic disease; or (iii) the interventions required surgery or medications. Study designs that were not RCTs and did not include measurement of the primary outcome of BMI/BMI z-score were also excluded. Due to lack of translation services, studies not reported in the English language were also excluded.

### 2.3. Study Selection

References and abstracts from all studies were downloaded to Endnote X8 citation management software (Thomson Reuters, Philadelphia, PA, USA). Duplicates were removed; abstracts and titles of the remaining articles were evaluated according to the eligibility criteria by two authors. Those that met the criteria had the full text retrieved, then screened by another independent reviewer. If any conflict arose on the eligibility of a study, a third author provided the final judgement.

### 2.4. Data Extraction

A data extraction template was made informed by the Cochrane data extraction guide and Preferred Reporting Items for Systematic Reviews and Meta-analysis (PRISMA) statement for reporting systematic reviews [18,21]. The template was piloted on selected studies and modified before being used to extract the following data: study details (authors, year and country of publication, affiliations and funding) participants (characteristics, recruitment, setting, attrition, compliance and blinding), randomization methods, primary outcome of BMI z-score or change in weight, secondary outcomes of changes in diet and physical activity and measurement of outcomes and analysis. Two authors independently completed data extraction. A third reviewer then independently extracted data and reviewed all three data extractions. Of the 29 emails sent to authors for effects of ethnic/racial group, only eight authors replied [22,23,24,25,26,27,28,29]. No further emails were sent to non-respondents.

### 2.5. Quality Assessment

The Cochrane assessment tool was used to assess the risk of bias at an individual level in RCTs [30]. The tool focused on biases in selection, attrition, detection and reporting. Two independent reviewers evaluated each study for biases as either low, medium or high risk. Conflicting decisions of risk rating were resolved by a third-party reviewer.

### 2.6. Grading of Recommendations Assessment, Development and Evaluation Assessment

The quality of the body of evidence was assessed by two reviewers using the Grading of Recommendations Assessment, Development and Evaluation system (GRADE) [31]. Five categories were assessed: limitations in study design and implementation; directness of evidence regarding study populations, study design and outcomes measured; inconsistency of results; precision of outcomes; and the probability of publication bias.

## 3. Results

### 3.1. Study Selection

The search yielded 11,366 records including eight additional articles from hand searches of paper’s reference lists. Duplications were removed, leaving 7409 abstracts for the first screening, thereafter another 7205 were excluded as they failed to meet the inclusion criteria, i.e., not an intervention, not the primary outcome, not the study design. Two hundred and four papers were screened for full text; 159 papers were excluded because they did not meet inclusion criteria. After the second screening, 30 studies from 45 papers [22,23,24,25,26,27,28,29,32,33,34,35,36,37,38,39,40,41,42,43,44,45,46,47,48,49,50,51,52,53,54,55,56,57,58,59,60,61,62,63,64,65,66,67,68] (including protocol and long term follow-up) were eligible to be included in this review. Of these studies, only three were conducted in ethnic/racial minorities [33,36,51] and four included analyses by ethnic and racial groups [28,29,61,68]. Figure 1 shows the review process in a Preferred Reporting Items for Systematic Reviews and Meta-Analyses(PRISMA) flowchart. Please see Appendix A for the list of full text articles excluded and supported reasoning.

### 3.2. Setting and Study Design

Of the 7 RCTs conducted in racial and ethnic minority groups, two were cluster RCTs [28,68]. Papers were published from 2006 to 2014. Five studies were performed in the United States of America (USA) [29,33,36,51,68], and two in the Netherlands [28,61]. Three interventions were conducted in schools [28,61,68], four were carried out in the community [29,33,36,51], of which two were based in the household [29,33]. Five were technology-mediated [28,36,51,61,68], of which one was solely a mobile technology intervention [51] and one study used a hand-held computer device [61]. The remaining three employed internet delivery [28,36,68].

Of the remaining 23 RCTs, 15 were cluster RCTs. All papers were published from 2006 to 2018. Six were conducted in Australia [22,41,45,47,53,64], followed by five studies conducted in the USA [24,25,27,52,55], followed by two studies in India [23,62], Italy [32,67] and Brazil [26,38], and one each in Iran [54], Greece [49], France [34], Sweden [44], the Netherlands [57], and Belgium [40]. Nineteen interventions were conducted in schools [22,23,24,25,32,35,38,40,41,43,44,46,48,53,54,57,62,64,67], four were carried out in the community [27,52,55,57] of which three were based in households [27,55,57]. Seven were technology-mediated [25,26,40,41,47,57,64] with one almost exclusively delivered online [57]. All papers reported sources of funding and one reported private sources of funding but remained undisclosed [34].

### 3.3. Study Characteristics

The duration of the seven interventions in minority groups ranged from 8 weeks [36] to 8 months [61]. Follow-up duration ranged from 6 [36] to 24 months [28] from baseline. Three did not conduct follow-up measures [29,51,68]. Two studies were single component dietary interventions [29,51], six were multicomponent with four deploying a combination of dietary and physical strategies [28,33,36,68] only to prevent overweight and obesity in adolescents from ethnic and racial minority backgrounds. All were educational and six were behaviour-change-based interventions [28,33,36,51,61,68]. Five were reported to be underpinned by an explicit theoretical framework(s) such as the following: Social Cognitive Theory [33,36], Social Learning Theory [68], Theory of Interactive Technology [68], Theory of Planned Behaviour [28], Trans Theoretical Model [36], Intervention Mapping [61] and the Precaution Adoption Process Model [28]. One study applied Motivational Interviewing [33], and two studies incorporated parental involvement [29,36] to aid adherence to the intervention and ultimately a healthier lifestyle for the long term. One intervention was grounded in weight control principles [51]. Lastly, all studies had one intervention and one control arm, with the control group not being exposed to the intervention.

The duration of the remaining 23 interventions ranged from nine weeks to two years. Only 10 had conducted follow-up measures [24,25,38,43,47,49,52,53,63,67], which ranged from 15 days to two years from baseline. Thirteen interventions did not conduct follow-up measures [22,23,27,32,34,35,40,41,44,45,48,55,57,58,62]. Of the 23 interventions, thirteen were multicomponent [23,24,25,26,34,38,40,41,47,52,53,55,62], and ten were single component, of which four were diet [32,49,54,67], five were physical activity-based [22,44,45,57,64], and one an environmental intervention [27]. Of the thirteen multicomponent interventions, nine were lifestyle-based, delivering a combination of dietary and physical activity to facilitate behaviour change [23,24,25,26,38,42,43,46,47,52,53,55]. Thirteen interventions were underpinned by one or more theoretical frameworks [24,25,26,37,38,40,41,42,43,44,45,46,47,49,52,53,54,63,64], with eight based on the Social Cognitive Theory [24,26,37,38,41,42,43,45,46,47,53,63,64], three on the Trans Theoretical Model [24,40,52], and one each on Social Determination Theory [63,64], Social Ecological Theory [41], Theory of Planned Behaviour [40], Health Promotion [44] and the Health Based Model [54], and seven interventions were considered behaviour-change-based [27,32,34,35,55,62,67]. Two incorporated Motivational Interviewing via individual counselling sessions for participants [24,38]. Fifteen involved parents to various extents to support participants in their implementation of healthy lifestyle behaviours [23,24,25,26,34,35,40,41,42,43,46,47,49,52,53,54,57,58,62,63,64]. All studies had two arms except the study by Bonsergent et al. [34], which had eight different groups sporting different combinations of educational, environmental and screening strategies.

### 3.4. Participants’ Characteristics and Recruitment Strategies

Participant characteristics for studies that tested on efficacy across ethnic and racial backgrounds were as follows; mean age of participants was from 11.3 to 15.3 years old, with an overall range from 11 to 17 years. All studies except one [51] recruited both sexes, with the majority being female and the remaining study recruiting females only. Of the seven studies, six reported on ethnic/racial group of its participants. The study by Singh et al. [61] did not report, as they found no interaction of ethnic/racial group with intervention effect and hence did not provide stratified analysis. Of the seven, two were conducted exclusively in a singular racial minority; one recruited from African Americans [33] and the other from American Chinese backgrounds [36]. Four [28,29,61,68] were universal interventions that included both White and minority populations [28,29,68], and one study recruited participants from various ethnic groups only [51]. The studies by Black et al. [33] and Nollen et al. [51] targeted adolescents from low-income neighbourhoods to access those from racial and ethnic backgrounds. Three recruited from schools [28,61,68], three from community and one intervention recruited from an existing longitudinal study and schools [33]. One study did not report recruitment methods [33]. Please see Table 1a for further information.

Participants’ characteristics for the remaining studies were as follows; mean age of participants within the studies was 11.3 to 15.8 years old, with overall range 11 to 17 years. Seven recruited only females [24,26,38,44,47,53,54], three recruited only males [45,53,64] and one did not report sex [27]. The remaining 12 studies recruited indiscriminately with the majority recruiting more females than males. Ten of the 23 studies reported ethnic/racial background of its participants [24,25,27,32,41,42,52,55,57,64] with two only reporting the proportion of participants from Caucasian background [27,57]. One study recruited only white [32]. Three studies had a diverse ethnic demographic [24,25,52,57,58,68,69] and six studies had mostly white participants [26,27,28,48,57,66]. One study targeted suburban areas for recruitment for their diverse student bodies [24]. Nineteen studies recruited participants from schools, four other studies recruited from within the community [27,52,55,57], of which one study recruited from both within the community and within schools [55]. Four did not report recruitment strategy [29,34,35,40,52]. Please see Table 1b for further information.

### 3.5. Primary and Secondary Outcomes; General and Targeted

Only seven studies reported on primary outcomes in racial/ethnic minority groups. All reported on BMI except one which used BMI z-score [33] as the primary indicator. One study [68] did not report mean difference or effect size. All studies failed to detect a significant difference in the primary outcome of BMI or BMI z-score between the intervention and control group [28,29,33,36,51,61,68]. Mean differences in BMI ranged from −0.0 [61] to −0.14 [29] kg/m^2^. The mean difference in BMI z-score was −0.03 [33] and another study reported β = 0.14 [28] for BMI [28]. One study reported a significant BMI difference of −0.75 kg/m^2^ (*p* = 0.03) for participants in the upper BMI tertile [29]. It is important to note that statistical significance was achieved for other obesity measures such as a decline (−0.25, *p* = 0.006) in the prevalence of overweight and obesity [33], waist to hip ratio (0.01, *p* = 0.02) [36] and bicep skin fold thickness for boys (−0.1) and girls (−0.3) [61].

Conversely, five of the seven studies detected changes in physical activity and dietary behaviours [28,29,33,36,61]. Three showed a significant decrease in Sugar Sweetened Beverages (SSB) consumption [28,29,61] with (OR = 0.54, 95% CI 0.34 to 0.88) [28], −249 mL/d [61] and −1201 kJ [29]. Also achieved was a reduction in snacks (β = −0.81) [28], in desserts (β = −2.21, *p* = 0.001) [33] and junk food [68]. Changes were also reported for increased fibre (β = −4.37, *p* = 0.036) [33], fruit (β = 0.41, *p* = 0.021) [33], fruit intake for at-risk students (β = 0.39) [28], vegetables (β = 19.3) [28] and combined fruit and vegetable intake [36,68], (0.14, *p* = 0.001) [36]. Meaningful decreases in sedentary behaviours were also reached in two studies [28,68] with one significant in males only (−25 min/wk) [28]. Two studies increased active behaviours, one in physical activity (12.46, *p* = 0.001) [36] and another in vigorous exercise [68]. No differences in intervention effect on primary and secondary outcomes by ethnic/racial group were found. None, however, reported on the effects of ethnic or racial group on attrition. Please see Table 2a for further information.

Of the 23 studies, seventeen found no significant difference in primary outcome of BMI or BMI z-score between intervention and control group. All six studies that were successful in establishing a difference in BMI/BMI z-score were carried out for both genders [25,34,40,41,49,67]. Of the six, four conducted subgroup analyses by gender [40,41,49,67], two showed no effect [49,67], one resulted in more weight loss in boys [41], and the other in girls [40]. Eleven reported on BMI z-score [26,27,32,34,40,41,43,45,47,52,55,67] and the remaining reported on BMI or both. Ten interventions did not report on a mean difference or effect size [22,32,38,40,44,49,52,54,62,67]. Mean differences in BMI ranged from 0.03 [34,35] to 0.72 kg/m^2^, mean differences in BMI z-score ranged from 0.004–0.09 [34,35] and another study reported effect size of 0.05 [53]. One intervention delivered an inverse effect with the control group decreasing significantly much more in the intervention group in BMI-SDS (β = 0.074, 95%CI: 0.008;0.14) and in skinfold thickness (β = 3.22, 95%CI: 0.27,6.17) [57,58]. Of the 23 studies, four studies also demonstrated changes in waist circumference [26,42,43,45,48,53,62], one in body fat (−1.96, *p* = 0.006) [46,47] and another in lean mass [22]. Four studies demonstrated significant difference in prevalence of obesity [25,33,34,55], of which two [33,55] did not exhibit significant difference in the primary outcome of BMI or BMI z-score.

Twenty studies reported secondary outcomes [22,23,24,25,26,27,29,32,42,44,45,47,49,52,53,55,57,62,64,67], with 14 of them having significant changes in either diet, physical activity or sedentary time. None of these studies carried out a subgroup analysis to determine the effect of ethnic/racial background on primary or secondary outcomes. Lastly, of the 23 studies, only two studies reported as to whether ethnic/racial background moderated attrition and both reported that there was no significant effect on the completion of the intervention [52,57]. Please see Table 2b for further information.

### 3.6. Risk of Bias

Using the Cochrane Risk of Bias tool, four of the seven RCTs conducted in ethnic minorities were rated as unclear risk of selection bias for randomization, as they did not specify method for random sequence generation [29,33,51,68]. Two of the seven studies reported allocation concealment for selection bias [28,29]. All seven studies were rated as low risk for attrition bias [28,29,33,36,61,68]. More than half (*n* = 4) of the studies were scored as unclear risk for detection bias due to insufficient information [29,36,51,68], and two were rated high as the researchers and research assistants were not blinded to group allocation [28,61]. One study was classified as low risk, as research assistants were blinded to group allocation and baseline findings [33]. Half of the studies (*n* = 3) were classified as having a low risk of reporting bias [28,61,68]. Please see Appendix A for further information.

Of the 23 RCTs, more than half (*n* = 14) did not specify method for random sequence generation and were rated as unclear risk of selection bias [23,24,27,32,34,35,37,40,44,45,46,47,48,52,54,55,62]. Only seven of 23 studies reported allocation concealment for selection bias [22,25,26,38,41,42,43,47,64]. Six studies did not analyse by intention to treat and thus had a high risk of attrition bias [24,32,40,44,45,48,55]. Two studies did not provide sufficient information on attrition and therefore classified as unclear risk of bias [25,40]. Eleven studies did not provide sufficient information on blinding and were rated for unclear risk of detection bias [23,32,34,35,40,41,52,54,55,62,64,67]. Four were of low risk [26,33,38,53]. Only nine studies were classified as low risk of reporting bias [25,26,28,34,35,38,41,45,47,57,61,64,68], while the remaining studies did not provide information on trial registration or publish a protocol and thus were classified as unclear. Please see Appendix A for further information.

### 3.7. Grading of Recommendations Assessment, Development and Evaluation Quality Rating

Of the 30 studies identified, seven studies reported findings for ethnic minorities for intervention effects on primary and secondary outcomes [28,29,33,36,51,61,68]. Thus, the GRADE tool was only applied to these studies to address the research question. These included a total number of 2763 adolescents. Please see Table 3 for further information.

#### 3.7.1. Study Limitations

Four of the seven studies did not state method of randomization [29,33,51,68] and only two studies reported allocation concealment method [28,29]. All used intention to treat analysis. Four studies [29,36,51,68] did not state method of blinding of personnel or participants, and two were deemed high risk of bias [28,61]. Blinding was low risk in only one study [33] and only three had a low risk of reporting bias [28,61,68].

#### 3.7.2. Consistency

Of the seven studies [28,29,33,36,51,61,68], those with universal recruitment [28,29,61,68] were unable to demonstrate a change in BMI outcome. Of the three that primarily targeted an ethnic minority [33,36,51], none were successful in showing a change for BMI and BMI z-score.

#### 3.7.3. Directness

Only seven of the thirty studies formed the evidence base [28,29,33,36,51,61,68] with an additional ten studies [24,25,27,32,41,42,52,55,57,64] capturing participant ethnicity without subgroup analyses of the primary outcome of change in BMI or BMI Z-Score. All interventions directly reported on the outcome of interest as they were all interventions aimed at preventing weight gain.

#### 3.7.4. Precision

The number of participants (*n* = 2763) across the seven studies [28,29,33,36,51,61,68] was less than the recommended 3000 to make definitive conclusions.

#### 3.7.5. Publication Bias

Efforts were made in ensuring all papers were captured, including an extensive search through seven major databases, hand searches of references list and contacting authors for additional information.

## 4. Discussion

To our knowledge, this is the first systematic review to investigate the efficacy of lifestyle interventions for the prevention of harmful weight gain in adolescents from ethnic and racial minority backgrounds. Analysis of the literature has revealed that despite their heightened vulnerability to overweight and obesity, there remains a dearth of lifestyle interventions that recruit and report on effectiveness in this priority population. Of the 30 studies captured in the search, only six were successful in preventing increases in BMI/BMI z-score [25,34,40,41,49,67]. Of the 30 studies reviewed, seven targeted ethnic minorities exclusively or reported subgroup analyses to determine if ethnic/racial minority status moderated intervention effect on BMI Z-score or BMI [28,29,33,36,51,61,68]. It is acknowledged that some studies may not have included subgroup analyses as sample power calculations did not allow for these comparisons. Furthermore, the small body and quality of evidence limits the interpretation and generalizability of results reported.

### 4.1. Effectiveness of Interventions in Preventing Harmful Weight Gain in Adolescents from Ethnic Minorities: Primary Outcomes

It is now well established that certain groups, such as those from ethnic and racial minority backgrounds, are disproportionately affected by obesity, and as such there has been a call for interventions to be targeted towards specific groups and be differentiated on factors of sex, age and SES [70]. Of the seven studies, no studies were successful in demonstrating a significant difference in BMI between the intervention and control group. Rather, most fell short of the BMI difference needed (0.57 [29]–0.8 [36] kg/m^2^) to reach the required effect size (e.g., 0.5) [29]. The greatest BMI change across the studies was 0.14 kg/m^2^, less than the 0.15 BMI effect considered to be clinically meaningful by Waters et al. [71]. Two web-based interventions offered the short follow-up periods (6 months from baseline) as a possible explanation for the lack of change in BMI [36,68]; however the small body of evidence limits any conclusions on intervention length to be drawn and the effectiveness of web-based interventions for ethnic and racial minority groups to remain unclear. Another possible explanation for the lack of efficacy is that, despite all four studies [28,29,61,68] including adolescents from ethnic/racial backgrounds, they did not incorporate co-design or consider ethnic minorities in the design of the universal interventions limiting influence and championship of the programs. They also did not employ specific recruitment strategies that targeted ethnic and racial minorities. It should also be considered that of the 23 interventions included in the review that did not consider ethnic/racial minority background, only seven were successful in preventing increases in BMI/BMI z-score [25,34,40,41,45,49,67].

An additional two of the seven studies conducted in ethnic/racial minority groups demonstrated some positive impact. While both studies were conducted in the community, the limited number prevents inferences from being made. The study of Black et al. [33], which targeted African-Americans exclusively, showed a reduced prevalence of overweight and obesity among the target population from 54% to 36% and from 32% to 34% in the intervention and control group respectively (−0.25 (0.09), *p* = 0.006). The absence of a significant change in BMI z-score was attributed to the inclusion of adolescents across a wide BMI range. Nevertheless, the intervention was still considered effective as it halted an increase in BMI category, with evidence also indicating that a 1% reduction in the prevalence of overweight and obesity in 16–17-year-old adolescents today has been projected to reduce the number of obese adults by 52,821 in the future, decrease lifetime medical costs by $586.3 million, and increase of quality-adjusted life years by 47,138 [41,72]. The intervention of Ebbeling et al. [29] was successful in decreasing BMI in those in the upper baseline-BMI tertile, which is the most important group to target. A subgroup analysis showed that this was consistent for all ethnic/racial subgroups. Similarly, there have been suggestions that while population-based primary prevention interventions should persist to target all children, the study aim and primary outcomes should be evaluated in the highest risk subgroup as opposed to the cohort at large [73].

### 4.2. Effectiveness of Interventions in Preventing Harmful Weight Gain in Adolescents from Ethnic Minorities: Secondary Outcomes

Five of the seven studies [28,29,33,36,39,61,74] demonstrated improvements in diet, physical activity or sedentary time, and this was similar to the ethnic groups. These included decreases in consumption of SSB [28,29,36,61,74] and snack foods, and increases in healthy foods such as fruit and vegetable intake [28,36], more physical activity [36] and reduction of screen time [36].

The findings were similar for the studies that recruited universal populations, with twenty showing significant improvements in diet, physical activity, or sedentary time [23,24,25,26,27,28,29,32,33,36,42,47,49,52,55,57,61,62,64,67,75]. This confirms previous findings from school-based interventions demonstrating effectiveness in the promotion of weight related attitudes, knowledge and behaviours such as healthy foods, physical activity and nutrition knowledge [76,77], but not change in weight status [19,78,79,80,81,82,83]. In our review, seven of the eight interventions carried out in the home or community setting also demonstrated improvements of lifestyle factors [27,29,33,36,52,55,57].

### 4.3. Setting: School or Community?

Traditionally, schools have been the predominant choice of setting for the delivery of interventions. This is expected, as schools offer ready and continuous access to adolescents as well as resources such as school policies, necessary personnel, curriculum, staff and facilities to promote physical activity and healthy eating [84,85].

Three [28,61,68] of the studies examining ethnic minorities [28,29,33,36,51,61,68] were conducted in the school setting, but none were effective in changing primary outcome of BMI, but changes in SSB consumption were reported [28,61]. Two of the three studies that were conducted in the community demonstrated positive impacts with one demonstrating significant changes in BMI for the most overweight/obese of the group and another reduced the prevalence of overweight and obesity. Community interventions support a family based approach by enabling parents to positively impact diet and physical activity habits [86].

A novel study by Chen et al. [36,74] was the first to explore the feasibility of a culturally specific, family based program delivered online for at-risk adolescents from an ethnic background in primary care clinics. This intervention was able to demonstrate significant changes in BMI and secondary outcomes and have them maintained at follow-up. The combination of a community setting and an online delivery allowed this intervention to leverage the strengths of each mode, to reach and engage this at-risk audience. The community setting allowed the early involvement of key stakeholders, with adolescents influencing the design and implementation of the intervention. The locally engineered nature of this intervention allowed the online and mobile technologies to be capitalised upon, as adolescents could choose from a variety of online learning methods, which is not possible in traditional face to face interventions. The use of mobile and online delivery might have increased equitable accessibility of the program with 95% of adolescents reporting they have access or own a smartphone [87]. A previous systematic review has highlighted the efficacy of this technology in increasing engagement in weight interventions and in decreasing adolescents’ dropout rate, which could have mitigated the often-reported difficulty in retaining and recruiting overweight youth in community-based programs [88,89].

### 4.4. Indigenous and First Nations

Of the six interventions carried out in Australia [22,41,45,47,53,64], only two reported ethnicity of participants [41,64] and only one study reported and recruited participants from Aboriginal and Torres Strait Islander background [41]. This study by Hollis et al. [41] was also the only intervention of the six that was able to demonstrate significant changes in BMI. However, no subgroup analyses were conducted to determine if the result was equally efficacious for all subgroups. Only five other interventions recruited and reported on First Nation Populations such as American Native Indians and African Americans [24,25,26,52,55], and similarly, only one study by Melnyk et al. was effective in demonstrating changes in BMI but did not test for differences [25]. The studies by Melnyk et al. [25] and Hollis et al. [41] were both multicomponent interventions underpinned by a theoretical framework, involved parents and employed behaviour change strategies including goal setting, social support, feedback and monitoring, identification and demonstration of the behaviour and used antecedents. These are only two studies of the six studies that impacted on BMI [25,34,40,41,49,67], of the thirty studies included in the review. This small body of evidence and lack of subgroup analysis prevents any meaningful conclusions to be drawn.

The mismatch between the evidence available to inform policy makers on how to intervene in this priority population and the existing disparities prevalent between First Nations and White populations is severe. In 2012–2013, more than a quarter of Indigenous people (37.4%) aged 10–14 and 34.9% of those aged 15–17 were overweight and obese, more likely than their non-Indigenous counterparts [90]. This is alarming considering that obesity was identified to be the second main contributor to the health gap between Indigenous and non-Indigenous people in Australia [91]. Furthermore, the intersection of socio-economic status and ethnicity/race should also be highlighted given that ethnic minorities tend to dominate the lower socio-economic bracket. For example, American Indians and Alaskan Natives have identified to be a racial group with the highest poverty rate in the U.S. [92] and Hispanics to be identified as an ethnic group with a poverty rate of 16% compared to the 8% experienced by the White population [93]. This was evident in selected studies choosing to recruit from low-income and suburban neighbourhoods due to their higher proportion of residents from ethnic and racial backgrounds [24,33,51]. Obesity prevention efforts for Aboriginal communities needs to be nested within the context of their history of colonisation, the major factor contributing towards their poor nutrition and health with the removal of traditional lands (and food sources) and prolonged financial stress associated with food insecurity which among other social inequalities has pressed Aboriginal communities to shift from traditional foods and consume the energy-dense Western diet [94]. Obesity intervention efforts focussed on adolescents from first nations should be at the forefront of prevention intervention priorities and should promote co-design, championship and governance with the First Nation population [95]. Of the thirty interventions, only one study conducted co-design with its participants from ethnic and racial backgrounds [50,51].

### 4.5. Study Strengths and Limitations

The lack of a standardized definition of ethnic/racial background and its indicators makes analysis difficult. Of the 30 studies, only five reported indigenous participants [24,25,26,41,51] and three studies divided race and ethnicity [29,51,69]. Concepts such as ‘race’ and ‘ethnicity’ are persisting issues in research [96] due to inconsistent definitions, lack of transparency in methodology employed to measure these concepts, and inappropriate classification of ethnicity and race. Similarly, variation in the use and combination of obesity indicators meant some studies were deemed unsuccessful despite eliciting some significant changes in other outcomes such as skinfold thickness [60,61], abdominal adiposity [62], waist circumference [26,42,43,62], waist hip ratio [62], and an increase in the maintenance and reduction of BMI for age respectively [55]. These limitations have been highlighted previously in a similar review among young adults from ethnic minorities [97].

### 4.6. Review Strengths and Limitations

A strength of the current review is the use of RCTs, which also ensures a higher level of evidence and contacting authors via email, therefore more information was obtained than was originally published. Among the limitations are that only interventions that were published in the English language were included. It is possible some successful interventions for ethnic minorities were not identified. Another strength was that efforts were made in ensuring all papers were captured, including an extensive search through seven major databases, hand searches of references list and contacting authors for additional information to minimise chances of publication bias.

## 5. Conclusions

Childhood programs for the prevention of harmful weight gain are important and interventions among adolescents are central to tackling the current obesity epidemic and halting its progression and its co-morbidities into adulthood. This review emphasizes that despite the need to focus efforts on priority populations such as those from ethnic/racial minorities and First Nations populations, to produce a meaningful decrease in the overall prevalence of overweight and obesity, there is an absence of studies. In effect, the review establishes the need for researchers to; actively engage and recruit from priority populations such as ethnic/racial minorities and to consider minority groups in the design and analysis of universal interventions.

## Figures and Tables

**Figure 1 ijerph-17-06059-f001:**
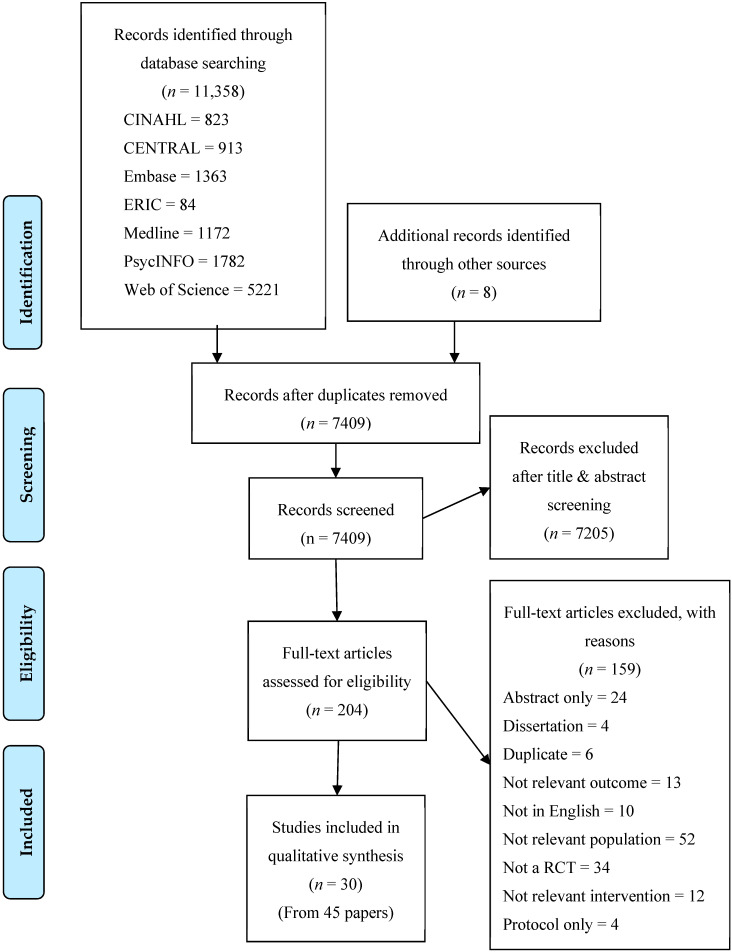
Preferred Reporting Items for Systematic Reviews and Meta-Analyses (PRISMA) flow chart for the search and filtering results for a systematic review of the effectiveness of prevention interventions for adolescents from racial and ethnic minorities.

**Table 1 ijerph-17-06059-t001:** Study and Participant Characteristics of interventions in the systematic review of effectiveness of prevention interventions for adolescents from ethnic/racial minorities.

First Author, Year, Country, Citation	Duration & Follow-Up	Study Characteristics	Participant Characteristics				Recruitment Methods	Funding
Study Design, Setting		Intervention Description/Comparator Description	*n*	Age	Ethnic/Racial Group	Sex		
**(a) Study and Participant Characteristics of interventions in the systematic review of effectiveness of prevention interventions for adolescents from ethnic/racial minorities (n = 7).**								
**Multicomponent Interventions**								
Singh et al. 2006 [59], 2007 [60], 2009 [61] Netherlands RCT School	8 mo (12, 20 mo)	Diet & PA, Environmental, Intervention mapping, Education, behaviour-change. I: 11 sessions on energy-related behaviours and reducing SSB, SB, High fat snacks + increasing active transport and sports. Individually computer-tailored advice, diary, pedometers + supportive video material. C: regular curriculum.	1053, I:632C:476	M: C: 12.9 ± 0.5 SD), I: 12.8 ± 0.5 (SD), F: C: 12.7 ± 0.5 (SD), I: 12.6 ± 0.5 (SD)	Ethnicity was tested in the regression model to determine any intervention but no stratified results shown.	F, M (50%)	Universal	Netherlands Heart Foundation, Ministry of VWS, Royal Association of Teachers of Physical Education
Black et al. 2010 [33] USA RCT Community	12 wks (24 mo)	Diet & PA, Education, SCT & MI I: 12 sessions with mentees. D and PA goals setting, tracking and evaluation. Healthy food testing + PA activity. C: No intervention	235, I:121, C:114	13.3 (11–16)	African American	I: F, M (48.8%), C: F, M (52.6%)	Targeted African American adolescents from low-income urban communities.	US Department of HHS, NCRR
Chen et al. 2011 [36] USA RCT Community (Internet)	8 wks (6 mo)	Diet & PA, Educational, TTM, SCT, Parental Involvement I: 8 sessions teaching participants + parents on emotions, goals and self-efficacy for a healthy lifestyle. Culturally appropriate. C: Received non-tailored general health info from website	54 I: 27, C: 27	12.52 ± 3.15 (SD)	NR	I: F, M (41%), C: F, M (52%)	Universal	NCRR, Hellman research grant, NIH
Ezendam et al. 2012 [28] Netherlands Cluster RCT School (Internet)	10 wks (4, 24 mo)	Diet & PA, TPB, Precaution Adoption Process Model, Implementation Intentions, Education I: On healthy eating, reducing SB, increasing PA. Tailored feedback on behaviour + determinants to prompt goal setting and action planning. C: regular curriculum.	883, I:485 C:398	12–13	I: Western 66%, Non-Western 34%, C: Western 78.9%, Non-Western 21.1%,	I: F, M (58.6%) C: F, M (49.7%)	Universal	ZonMw (The Netherlands Organisation for Health Care Research and Development)
Whittemore et al.2013 [68] USA Cluster RCT School (Internet)	6 mo	Diet & PA, SLT, Theory of Interactive Technology I: HeT program +stress reduction, assertive communication, conflict resolution and social problem skills) C: HeT program on nutrition, portion control, PA and metabolism. Individualized feedback and goal setting, encouraged self-monitoring of food intake and PA and opportunity to interact with health coach.	384	15.31 ± 0.69 (SD)	65% non-white	F, M (38%)	Universal	Jonas Centers for Nursing Excellence, NINR, NIH,
**Single Component Dietary Interventions**								
Ebbeling et al. 2006 [29] USA RCT Community	25 wks	Diet, Environmental, Parental Involvement I: Home deliveries of non-caloric drinks. Magnets with possible side effects of SSBs mailed monthly. No SSB permitted. Telephone calls to participant + parent to reinforce and motivate. C: Continue as normal. Weekly non-caloric beverage delivery after completion of FU as incentive.	103, I:53, C:50	15.9 ± 1.1 (SD)	White: C: 56%, I: 55%, Black: C: 24%, I: 24%, Asian: C: 4%, I: 4%, Multiple or other: C:17%, I: 18%, Hispanic: C: 17%, I: 25%, Non-Hispanic: C: 83%, I:75%	F, M (45.6%)	NR	None
Nollen et al. 2012 [50], 2014 [51], USA RCT Community (Internet)	12 wks	D, Screen Time, Behaviour-based Both targeted F &V, SSB and Screen Time I: To set 2 daily goals + behaviour plan. Also prompted girls to self-monitor goal progression 5 times/day. Girls received 1 song/day if they responded to 80% of daily prompts C: Received manuals (snapshots of respective module) at wks 1, 5 and 9 only.	51	11.3 ± 1.6 (SD)	Hispanic/Latina: total 7.8, Race: African American: 83.7 Bi- or Multi-racial: 8.2, American Indian/Alaska Native: 6.1, Asian/Pacific Islander: 2.0	100%F	Targeted: girls from racial and ethnic minorities from low income neighbourhoods	ORWH, NICHD, NIAID, NIMH, NHLBI
**(b) Study and Participant Characteristics of interventions in the systematic review of effectiveness of prevention interventions for adolescents from ethnic/racial minorities (n = 24).**								
**Multicomponent Interventions**								
Briancon et al. 2010 [35], France Bonsergent et al. 2013 [34], France RCT School	24 wks	Education, Environmental, Screening, Parental Involvement 2 × 2 × 2 design, 3 Intervention types Education: Nutrition and PA lectures, problem solving re PA, eating habits and the environment. End of year parties for reinforcement. Environment: Increased availability of water, F & V, dairy, bread and PA in schools. Posters/signs promote changes. Screening: Students assessed against anthropometrics and psychological variables, at-risk students referred to care management implemented by external nutrition network. C: No intervention	3538	15.8 ± 0.02 (SE)	NR	Sex: F, M (47.1%)	NR	Grants from Private and Public Sectors
Haerens et al. 2006 [40], Belgium Cluster-RCT School	12 mo (24 mo)	Education, Environmental, TTM, TPB, Parental Involvement I: (Classroom lessons + individual computer tailored intervention for PA, fat and fruit + PA sessions and cheap/free fruit wkly. Schools received sports materials + free water cans.) I + P: Included Parental Involvement. Interactive meeting on PA, healthy food, obesity and health, newsletters, CD-ROM with computer intervention on fat intake and PA. C: NR		13.06 ± 0.81 (SD)	NR	F, M (63.4%)	NR	Policy Research Centre Sport, Physical Activity, and Health funded by the Flemish Government
Hollis et al. 2016 [41], Australia Sutherland et al. 2013 [65],2016 [66] Australia Cluster-RCT Schools	19–24 mo	PA, Education, Environmental, Socio-ecological theory, SCT Parental Involvement I: Enhanced school sports program, strategies to increase PA in PE classes + school breaks + school policy changes, and parental + community engagement. C: Requested to follow usual PE and sports programs.	1150	11–13	Aboriginal and Torres Strait Islander, C: 8.8%, I: 8.4%	F, M (49%)	Universal	NSW Ministry of Health
Singhal et al. 2010 [62] India RCT School	6 mo	Educational, Diet & PA, Environmental, Parental Involvement I: Lectures + activities to promote PA, diet, healthy lifestyle. Individual counselling, school policy changes, health camp with parents, parent counselling, training student volunteers C: No intervention	201	15–17	NR	F, M (60%)	Universal	World Diabetes Foundation
Thakur et al. 2016 [23], India Cluster-RCT School	20 wks	Educational, Diet & PA, Environmental, Parental Involvement I: Diet and PA, environment, and lifestyle disorders. Mandatory inclusion one period of PA/day in school, healthy school canteen. Parents made Diet recommendations + reducing screen time. C: Diet and PA info if desired	462	13.5 ± 0.7 (SD)	NR	F, M (81%)	Universal	Indian Council of Medical Research
Dunker et al. 2017 [38], Brazil Cluster-RCT School (Phone)	9 wks (18 wks)	Education, PA, Diet, SCT, MI I: after-school PA and education. Individual counselling, didactic resources + lunch on days of activities C: No intervention	270	13.4 ± 0.64 (SD)	NR	100% F	Universal	Sao Paulo Research Foundation, CNPq
Leme et al. 2015 [43], 2016 [26], 2018 [42] Brazil Cluster-RCT School (Phone)	6 mo (6 mo)	Diet & PA, Education, SCT, PI I: Cultural adaptation of NEAT Girls study, PA and low-cost healthy eating. C: No intervention	253	14–18	62.8%% White, 11.5% Afro descendent, 0.8% Asian, 24.1% Brown, 0.8% Native Indian	100% F	Universal	FAPESP, federal funds from USDA ARS
Lubans et al. 2010 [46],2012 [47] Australia Dewar et al. 2013 [37], Australia Cluster-RCT School (Phone)	12 mo (2 yrs)	Diet & PA, Education, peer support, SCT, Parental Involvement I: Enhanced school sport & lunchtime PA, interactive educational seminars and nutrition workshops. Pedometers, handbooks for participants + parents and text prompt messages. Parents received termly newsletters. C: No intervention	357	13.18 ± 0.45 (SD)	NR	100% F	Universal	ARC Discovery Project Grant
Melnyk et al. 2013 [25], USA Cluster-RCT School	15 wks (6 mo)	Educational, PA, CT, Parental Involvement I: Health education course with COPE taught cognitive-behavioural skills and focused on PA and diet info + PA sessions, homework activities, parental newsletters. Pedometers to increase step count 10%/wk. C: Received Healthy Teens program. Safety + common health topics (e.g., road safety, skin care, dental care).	807	14.74 ± 0.73 (SD)	2.5% American Native, 4% Asian, 9.9% black, 14.1% White, 67.5% Hispanic, 1% other	F, M (48.4%)	Universal	NIH, NINR
Neumark et al. 2010 [24] USA Cluster-RCT, School	16 wks (9 mo)	Diet & PA, Education, SCT, TTM, Parental Involvement, MI I: New Moves curriculum during PE class. Nutrition education and social support/self-empowerment. Counselling sessions, lunch sessions and parent outreach activities. C: No intervention, told to conduct physical education classes as usual	356	15.8 ± 1.17 (SD)	Over 75% of the girls were racial/ethnic minorities: Black/African America: 28.4% White: 24.4% Asian: 23% Hispanic: 14.3% Mixed/Other: 7.3% American Indian: 2.5%	100% F	Targeted suburban areas for their diverse student bodies.	NIDDK, NIH
Patrick et al. 2006 [52] RCT Community (Primary Health Care Settings) (Phone)	12 mo (2 yrs)	Behaviour change, TTM, Parental Involvement I: Participants participated in PACE+. Computer nutrition assessment (fat intake, F & V intake) + PA behaviours + stage of change, then developed a tailored behaviour change Progress Plan for 1 nutrition and 1 PA behaviour. Printed guide + telephone counselling + mailed worksheets and tips C: Received SunSmart Protection program.Parents encouraged to support via praising, active support and role-modelling.	819	12.7 ± 1.3 (SD)	Asian or Pacific Islander: 3.2, African American: 6.6, Native American: 0.7, Hispanic: 13.1, White: 58.4, Multi-ethnic or other:18.0	F, M (47%)	NR	NIH, NCI Bethesda, Md.
Peralta et al. 2009 [53] Australia RCT School	16 wks (6 mo)	Educational, Diet & PA, SCT, Parental Involvement I: Received curriculum sessions on PA, SSB, ST, and increasing fruit consumption via increased self-efficacy. Practical components which promoted PA + parental newsletters. C: Regular PA sessions at same time.	33	12.5 ± 0.4	NR	M (100%)	Universal	Participating students, staff and broader intervention school community (partial)
Rodearmel et al. 2007 [55] USA RCT Community (household)	24 wks	Diet and PA, I: To increase daily PA by 2000 steps/day + reduce EI by 420 kJ/day with changing sugar for non-caloric sweeteners C: Families were asked to maintain, monitor, and report their usual lifestyle for the duration of the study. All SM family members were asked to wear pedometers	298	I: 11.11 ± 2.08 (SD) C:11.28 ± 2.29 (SD)	I: White: 52.59% Black: 13.79% Hispanic: 13.79% Other: 19.38% NR: 0.00% C: White: 50.98% Black: 18.63% Hispanic: 12.75% Other: 15.69% NR: 1.96%	C: F, M (46%) I: F, M (49%)	Universal	McNeal Nutritionals, LLC, NIH
**Single Component Dietary Interventions**								
Amaro et al. 2006 [32], Italy Cluster-RCT School	24 wks	Diet, Educational, Behaviour-change, I: Kaledo (board game) sessions re Mediterranean diet, energy intake, expenditure and balance. C: No Intervention	291	11–14	White	F, M (63%)	Universal	Italian Association Amici di Raoul Follereau, Commune of Naples, Second University of Naples
Mihas et al. 2009 [49], Greece Cluster-RCT School	12 wks; (15 days + 12 mo)	Diet, Social Learning Theory Model, Parental Involvement I: Workbook covering dietary issues + dental healthy hygiene + consumption attitudes. Classroom modules included health and nutrition education. Included 2 educational parent meetings. C: No health education intervention + no parental education. Medical screening results sent to parents	218	13.3 ± 0.9 (SD)	NR	C: F, M (49.5%) I: F, M (49%)	Universal	Ministry of Education, National Foundation for the Youth
Rabiei et al. 2017 [54], Iran RCT Schools	2 mo, 3 mo	Educational, Diet, HBM, Parental Involvement I: Lectures, Q and A, educational booklets and pamphlets. Lectures targeted perceived susceptibility, severity and self-efficacy. C: NR	140	NR	NR	100% F	Universal	Research Department of Isfahan University of Medical Sciences
Viggiano et al. 2015 [15], Italy Cluster-RCT School	20 wks (6 + 8 mo)	Education, Diet, Behaviour-based I: Play sessions involving Kaledo (as per Amaro et al. 2006). C: No play sessions with Kaledo	3110	9–19	NR	F, M (55%)	Universal	Second University of Naples, Sport, Kaledo Cultural Association, Campania Region (Department of Education), Naples, Salerno, Cercola, Department of Sport, Foundation for Child Care
**Single Component Environmental Interventions**								
French et al. 2011 [27], USA Cluster-RCT Community (household) (Phone)	1 yr	Environmental, Behaviour-based I: Group sessions, (time-limiting devices on TVs + home scale + guidelines for food availability), GS, positive reinforcement, self-monitoring), home activities + telephone support calls C: No intervention	90	12–17	79% White	NR	Universal	NIH/NCI
**Single Component Physical Activity Interventions**								
Lindgren et al. 2011 [44], Sweden Cluster RCT School	6 mo	PA, self-efficacy, Health Promotion I: Participants invited to master different exercise and sports activities in safe, non-judgmental environment with other non-active girls of similar age + discussion time (e.g., healthy lifestyles) C: No intervention	110	C: 15.5 ± 1.1 (SD), I: 15.3 ± 1.9 (SD)	NR	100% F	Universal	Halland Regional Development Council, The Primary Health Care Research and Development Unit, Halland County Council, Falkenberg, Sweden.
Lubans et al. 2011 [45], 2016 [48] Australia Cluster-RCT School	3 mo (6 mo)	Educational, PA, SCT I: Involved school sport and lunchtime PA sessions, interactive seminars, PA leadership + nutrition handbooks and pedometers for self-monitoring. C: No intervention	100	14.3 ± 0.6 (SD)	NR	100% M	Universal	HMRI, Rotary Club of Newcastle Enterprise
Simons et al. 2014 [58], 2015 [57] (Netherlands) RCT Community (Household) (Internet)	10 mo	PA, Parental Involvement I: Received a PlayStation Move + 5 active video games. Encouraged to substitute non-active with active gaming for at least 1 hr/week C: No intervention	270	13.9 ± 1.3 (SD)	White—83%	F, M (91%)	Universal	ZonMw—The Netherlands Organization for Health Research and Development
Smith et al. 2014 [63,64] Australia Cluster-RCT School (Phone)	20 wks (8 + 18 mo)	Educational, PA, SDT, SCT, Parental Involvement I: Educational seminar, enhanced school sport +, lunch-time PA mentoring sessions. Pedometer + smartphone app for self-monitoring. School exercise equipment pack + 4 Parental newsletters C: Usual practice, provided with condensed program after 18-mo assessments.	361	12.7 ± 0.5 (SD)	Australian 73.7%, European 17.3%, African 3.4%, Asian 1.7%, Middle Eastern 1.1% and other 2.8%.	100% M	Universal	ARC, NHMRC NHFA Career Development Fellowship
Weeks et al. [22] 2012 Australia RCT School	8 mo	PA I: 10 min of supervised jumping activities at beginning of each PE class C: Regular PE warm-ups and stretching directed by usual PE teacher	99	13.8 ± 0.4 (SD)	NR	F, M (46%)	Universal	No external funding sources

USA, United States of America; mo, months; wks, weeks; NS, not significant; PI, post-intervention; FU, follow-up; F &V, fruit and vegetable; PA, physical activity; SSB, sugar sweetened beverages; SB, sedentary behaviour; I, Internet; RCT, randomized controlled trial, C, control; I, intervention; NR, not reported; PE, Physical Education; SCT, social cognitive theory; MI, motivational interviewing; TTM, Trans theoretical Model; TPB, Theory of Planned Behaviour; FU, Follow-up; SLT, Social Learning Theory; EB, Eating Behaviour; JF, Junk Food; FF, Fast Food; VWS, Health, Welfare and Sport; NCRR, National Centre for Research Resources; HHS, Health and Human Services; NINR, National Institute of Nursing Research; NIH, National Institutes of Health; ORWH, Office of Research on Women’s Health; NICHD, The Eunice Kennedy Shriver National Institute of Child Health and Human Development; NIAID, National Institute of Allergy and Infectious Diseases; NIMH, National Institutes of Mental Health; NHLBI, National Heart, Lung, and Blood Institute; CT, Cognitive Theory; CNPq, Brazilian National Council for Scientific and Technological Development; ARC, Australian Research Council; NIDDK, National Institute of Diabetes and Digestive and Kidney Diseases; NIC, National Cancer Institute; HMRI, Hunter Medical Research Institute; NHFA, National Heart Foundation of Australia.

**Table 2 ijerph-17-06059-t002:** Study outcomes of interventions in the systematic review of effectiveness of prevention interventions for adolescents from ethnic/racial minorities.

**First Author, Year, Citation, Country**	**Outcomes of Intervention**		**Intervention Subgroup Analysis by Racial/Ethnic Minority Status**		**Attrition (%)**	**Attrition Subgroup Analysis by Racial/Ethnic Minority Status**
	**Primary**	**Secondary**	**Primary**	**Secondary**		
**(a) Study outcomes of interventions in the systematic review of effectiveness of prevention interventions for adolescents from ethnic/racial minorities (n = 7)**						
**Multicomponent Interventions**						
Singh et al. 2006 [59], 2007 [60], 2009 [61] Netherlands	F BMI ∆: −0.1 (−0.2 to 0.1) M BMI ∆: −0.0 (−0.1 to 0.2) Significant F BSF: −0.3 (−0.7 to 0.3) M BSF: −0.1 (−0.4 to 0.2)	Significant SSB F: −249 (−400 to −98), M: −287 (−527 to −47) Significant SB ∆: M at FU: −25 (−50.0 to −0.3) NS SB ∆: −22 (−55 to 2)	No effect	No effect	21	NR
Black et al. 2010 [33] USA	BMI z-score: −0.03 (0.06) SE (*p* = 0.574) Prevalence: −0.25 (0.09) (*p* = 0.006)	S & D: β = −2.21 (0.66) SE, (*p* = 0.001) β = −0.69 (0.31) SE (*p* = 0.026) at FU. Fibre: β = −4.37 (2.07) SE, (*p* = 0.036) F: β = 0.41 (0.18) SE(*p* = 0.021) PA: β = 10.76 (7.53) SE (*p* = 0.155) V: β = −0.18 (0.31) SE (*p* = 0.559) Milk: β = 0.13 (0.22) SE (*p* = 0.556) Non-diet soda: β = −0.04 (0.13) SE (*p* = 0.745) Fried foods: β = −0.08 (0.09) SE (*p* = 0.375) Calcium: β = 10.76 (7.53) SE (*p* = 0.155) Saturated Fat: β = −5.54 (3.37) SE (*p* = 0.102) Total Fat: β = 17.01 (9.28) SE (*p* = 0.069) Total energy: β = −459.73 (235.37) SE (*p* = 0.053)	NA (100% African American)	NA (100% African American)	23.8	NA (100% African American)
Chen et al. 2011 [36], USA	BMI: 0.01 (−0.3, 0.04), (*p* = 0.84) WHR: −0.01 (−0.01, −0.001), (*p* = 0.02)	F &V: 0.14 (0.06, 0.22) (*p* = 0.001) PA: 12.46 (6.62, 18.41) (*p* = 0.001)	NA (100% Chinese American)	NA (100% Chinese American)	8.4	NA (100% Chinese American)
Ezendam et al. 2012 [39] Netherlands	BMI: β = 0.14 (−0.17 to 0.45), WC: β = 0.60 (−0.44 to 1.64)	SSB (OR, 95% CI): 0.54 (0.34, 0.88) Snacks: β = −0.81 (−1.33, −0.29) V: β = 19.3 g/d (7.54, 31.21) At-risk students: F: β = 0.39 g/d (0.13, 0.66) Step Count: β = 14 228 steps/wk (678, 27,838) FU Whole Wheat Bread: OR 1.08 (0.67, 1.75) SB: β = −5.4 (−25.2, 14.5)	No effect	No effect	14	NR
Whittemore et al. 2013 [68] USA	BMI: I: 24.5 (5.4), 24.6 (5.4) C: 25.0 (5.7), 25.1 (5.6) (*p* = 0.87)	SB ∆: I: 5.6 (2.2), 5.3 (2.3), C: 5.4 (2.2), 5.2 (2.3) (*p* < 0.01) F&V: I: 4.9 (2.0), 5.1 (1.9), C: 5.0 (2.3), 4.9 (2.1) (*p* < 0.01) Total EB: I:56.8 (11.9), 56.4 (11.9), C: 56.6 (11.1), 57.2 (10.6) (*p* < 0.01) JF: I: 2.5 (2.2), 2.7 (2.4), C: 2.4 (2.0), 2.5 (1.9) (*p* < 0.01) VE: I: 4.1 (2.2), 4.1 (2.1), C: 3.7 (2.2), 4.1 (2.2) (*p* < 0.01) BF: I: 4.1 (2.6), 3.7 (2.7), C: 4.2 (2.4), 3.9 (2.5) (*p* = 0.9211) FF: I: 0.83 (1.09), 0.80 (1.03), C: 0.72 (0.91), 0.85 (1.00) (*p* = 0.0892)	No effect	No effect	4.9	NR
**Single Component Dietary Interventions**						
Ebbeling et al. 2006 [29] USA	BMI: (−0.14 ± 0.21 kg/m^2^). If baseline BMI ≥ 25.6 kg/m: −0.75 ± 0.34 kg/m^2,^ (*p* = 0.03)	SSB ∆: I: −1201± 836, C: −185 ± 945 (*p* < 0.001) Non caloric beverage ∆: I: 396 ± 493, C: 78 ± 523 (*p* = 0.002) PA∆: I: −0.12 ± 0.37, C: −0.03 ± 0.32 (*p* = 0.18) Television viewing∆: I: 0.05 ± 1.56, C: −0.19 ± 1.85 (*p* = 0.47) Total media time∆: I: −0.50 ± 2.56, C: −0.31 ± 3.33 (*p* = 0.75)	No effect	No effect	0	NR
Nollen et al. 2012 [50], 2014 [53] USA	ES = 0.03, (*p* = 0.91)	F&V: 0.44, (*p* = 0.13) SSB: −0.34 (*p* = 0.25) Screen time: 0.09 (*p* = 0.76)	No effect	No effect	13.7	NR
**First Author, Year, Citation, Country**	**Outcomes of Intervention**				**Attrition (%)**	**Attrition Subgroup Analysis by Racial/Ethnic Minority Status**
	**Primary**		**Secondary**			
**(b) Study outcomes of interventions in the systematic review of effectiveness of prevention interventions for adolescents from ethnic/racial minorities (n = 24)**						
**Multicomponent Interventions**						
Briancon et al. 2010 [35] France Bonsergent et al. 2013 [34] France	Education BMI ∆: 0.71 ± 1.49 (*p* < 0.0001) BMI z-Score ∆: −0.07 ± 0.44 (*p* < 0.0001) No Education BMI ∆: 0.66 ± 1.45 (*p* < 0.0001) BMI z-Score: −0.07 ± 0.43 (*p* < 0.0001) Education vs. No Education: BMI ∆: 0.05 (−0.05, 0.15) (*p* = 0.2858) BMI z-score ∆: 0.004 (−0.026, 0.034) (*p* = 0.8118) Environment: BMI ∆: 0.71 ± 1.47 (*p* < 0.0001) BMI Z-score ∆: −0.06 ± 0.44 (*p* < 0.0001) Non Environment BMI ∆: 0.67 ± 1.47 (*p* < 0.0001) BMI Z-score ∆: −0.07 ± 0.43 (*p* < 0.0001) Environment vs. non environment: BMI ∆: 0.03 (−0.07, 0.13) (*p* = 0.5028) BMI z-score ∆: 0.005 (−0.025, 0.035) (*p* = 0.7460) Screening BMI ∆: 0.64 ± 1.44 (*p* < 0.0001), BMI z-score ∆: −0.09 ± 0.44 (*p* < 0.0001) No Screening BMI ∆: 0.72 ± 1.49 (*p* < 0.001) No screening BMI z-score ∆: −0.05 ± 0.43 (*p* < 0.0001) Screening vs. No screening: BMI ∆: −0.11 (−0.21, −0.01) (*p* = 0.303) BMI Z-score ∆: −0.036 (−0.066, −0.007) (*p* = 0.0173)		*N/*A		55.5	NR
Haerens et al. 2006 [40] Belgium	M BMI: I + P: 19.24 ± 3.62, 19.79 ± 3.64, 20.52 ± 3.68 I: 19.32 ± 3.35, 19.98 ± 3.35, 20.86 ± 3.51 C: 18.58 ± 2.91, 18.99 ± 2.82, 19.67 ± 2.89 M BMI z-score: I + P: 0.07 ± 1.09, 0.17 ± 1.03, 0.16 ± 1.04 I: 0.10 ± 1.02, 0.22 ± 0.97, 0.25 ± 0.98 C: −0.07 ± 0.98, −0.02 ± 0.092, −0.04 ± 0.94 P = NS (NR) F BMI: I + P: 20.26 ± 3.95, 20.75 ±3.90, 21.34 ± 3.83 P = Significant (NR) I: 20.23 ± 3.60, 20.94 ± 3.54, 21.66 ± 3.68 C: 19.23 ± 3.52, 19.94 ± 3.65, 20.78 ± 3.66 P = NS (NR) F BMI z-score: I + P: 0.07 ± 1.09, 0.28 ± 1.08, 0.23 ± 1.12 P = Significant (NR) I: 0.09 ± 1.06, 0.39 ± 0.90, 0.27 ± 0.96 C: 0.07 ± 0.98, 0.11 ± 1.03, −0.01 ± 1.06 P = NS (NR)		*n*/A		NR	NR
Hollis et al. 2016 [41], Australia Sutherland et al. 2013 [65], 2016 [66] Australia	BMI ∆: −0.28 kg/m^2^ (−0.50; −0.06), (*p* = 0.01) −0.28 kg/m^2^ (−0.49; −0.06), (*p* = 0.01) FU BMI z-score ∆: −0.05 (−0.11; 0.01), (*p* = 0.13) −0.08 (−0.14; −0.02), (*p* = 0.02) FU Normal/Underweight: BMI ∆: −0.33 kg/m^2^ (−0.55; −0.10), (*p* = 0.01), BMI z-score ∆: −0.08 kg/m^2^ (−0.15; −0.01), (*p* = 0.01) FU Overweight BMI ∆: −0.39 kg/m^2^ (−1.01; 0.22), (*p* = 0.21) −0.18 kg/m^2^ (−0.80; 0.44), (*p* = 0.45) BMI z-score ∆: −0.07 (−0.21; 0.07), (*p* = 0.31) −0.00 (−0.14; 0.14), (*p* = 0.54).		*n*/A		8.6 (14.3)	NR
Singhal et al. 2010 [62] India	BMI ∆:95% CI (−0.18 to 0.34), (*p* = NR), WC ∆: −2.43 to −0.17 (*p* = 0.02)		Milk ∆: I: 32.8% (*p* < 0.001), C: 7.8 %, (*p* = 0.152 Whole pulses: I: 6.6 (*p* = 0.392), C:0.7 (*p* = 1) Sprouts (>2 times/wk), I: 3.7 % (0.648), C: 2.5% (*p* = 0.644), Nuts: I: 9.4 % (*p* = 0.286), C: 14.9 % (*p* = 0.111), Green leafy veggies: I: 7.5% (*p* = 0.349), C: 5.7% (*p* = 0.636), Fresh fruits: I: 9.9%, (*p* = 0.856), C: 6.5% (*p* = 0.268), White bread: I: 11%, (*p* = 0.004), C: 3.4% (*p* = 0.608) Biscuits: I: 7.9% (*p* = 0.430), C: 0.7% (*p* = 0.749), Aerated drinks: I: 15.1% (*p* = 0.001), C: 7.7% (*p* = 0.265), Aerated drink: I: 0.2%, (*p* = 1), C: 9.8% (*p* = 0.087), Western junk: I: 8.9%, (*p* = 0.031), C: 0.7% (*p* = 1) Chips: 7.8% (*p* = 0.152), C: 0.5% (*p* = 1), Indian junk: I:6% (*p* = 0.265), C: 0.8% (*p* = 1), PA: 2.4%, (*p* = 0.169), C: 5.9% (*p* = 0.377), PA (time): 9.8%, (*p* = 0.164), C: 3% (*p* = 0.755), Bring tiffin home, I: 14.9%, (*p* = 0.004), C:1.8%, (*p* = 0.263), Fruit in tiffin: I: 30.7% (*p* < 0.001), C: 3.9% (*p* = 0.585), Fruit in tiffin (>3 times/wk): 14.5%, (*p* = 0.001), C: 1% inc (*p* = 1) Household chores I: 8%, (*p* = 0.215), C: 2% (*p* = 0.839), Eating Out: I:1.7% (*p* = 0.143), C: 10.8% (*p* = 0.027). Eating out (canteen): I: 13.1%, (*p* = 0.001), C: No change, (*p* = 1), Watching TV: I: 4.9%, (*p* = 0.302), C: 3%, *p* = 0.629), Board game (sed activity): I:4%, (*p* = 0.503), C: 3%, (*p* = 0.607), Tuition classes (sed activity): I: 5% (*p* = 0.267), C: No change (*p* = 1)		3.8	NR
Thakur et al. 2016 [23] India	BMI ∆: −0.09 (−0.19 to 0.01) (*p* = 0.09)		Energy change: −0.18 (−0.34 to−0.02) (*p* = 0.02), Protein: −0.25 (−0.40 to −0.10) (*p* = 0.001), Fat: −0.30 (−0.47 to −0.13) (*p* = 0.01), Dietary fibre: −0.22 (−0.42 to v0.02) (*p* = 0.03) School related MET: −0.56 (−0.75 to −0.37) (*p* < 0.001), Transport related MET: 0.30 (0.12 to 0.48) (*p* = 0.001), Total METs score: 0.06 (−0.12 to 0.25) (*p* = 0.50)		19.3	NR
Dunker et al. 2017 [38] Brazil	BMI: I: 21.6 (95% CI 20.75, 22.45), C: 22.28 (95% CI 21.47, 23.1)		*n*/A		15.2	NR
Leme et al. 2015 [43], 2016 [26], 2018 [42] Brazil	BMI ∆:−0.26 kg/m^2^ (*p* = 0.08) BMI z-score ∆: −0.07 (*p* = 0.14) WC ∆: −2.28 cm (*p* = 0.01) Prevalence of Overweight: I (20.4% vs. 19%), C: (16.2% to 18%)		F: 0.26 (0.13) (*p* = 0.010), V: 1.16 (0.60) (*p* = 0.009) Sweets: −0.62 (0.39) (*p* = 0.109) Oils: −0.48 (0.39) (*p* = 0.229) Sedentary (wknd): −0.92 (0.35) (*p* = 0.005) Computer time (wknd): −0.63 (0.24) (*p* = 0.015)		24.9	NR
Lubans et al. 2010 [46], 2012 [47] Australia Dewar et al. 2013 [37] Australia	BMI ∆: −0.19 (−0.70 to 0.33), −0.33 (−0.97 to 0.28) (*p* = 0.353) FU BMI z score ∆: −0.08 (−0.20 to 0.04), −0.12 (−0.27, 0.04), (*p* = 0.178) FU Body Fat ∆: −1.96, (−3.02, −0.89) (*p* = 0.006)		NS		17.6 (33.6)	NR
Melnyk et al. 2013 [25] USA	BMI ∆: −0.20 (−0.35, −0.05) (*p* = 0.01), −0.34 (−0.56, −0.11) (*p* = 0.00) Proportion overweight: 0.45 (0.42, 50) (*p* = 0.03)		Steps/day ∆: 4061.83 (1437, 6686.66) (*p* = 0)		13.6 (22.3)	NR
Neumark et al. 2010 [24] USA	BMI: −0.08 (*p* = 0.512), −0.10 (*p* = 0.446) FU		PA: 0.08 (*p* = 0.894), 1.20, (*p* = 0.068) F&V: 0.24 (*p* = 0.365), SSB: −0.05 (*p* = 0.751) Sedentary activity: −0.12, (*p* = 0.834), −1.26 (*p* = 0.050), TV: 0.51 (*p* = 0.158), −0.05 (*p* = 0.883)		3.1 (5.6)	NR
Patrick et al. 2006 [52]	F: (*p* = 0.069), M: (*p* = 0.53)		Sedentary behaviours: F I: 4.3 to 3.4 h/d C:4.2 to 4.4 h/d (*p* = 0.001), M I: 4.2 to 3.2 h/d C:4.2 to 4.3 h/d (*p* = 0.001) M Active Time I: 4.1 to 4.4 d/wk C: 3.8 to 3.8 d/w (*p* = 0.01), F Fat intake: RR 1.33 (1.01–1.68) M PA: RR 1.47 (1.19–1.75)		7.3 (19.3)	No effect
Peralta et al. 2009 [53] Australia	BMI ∆: −0.2 (−0.8, 0.4) ES = 0.05 (*p* = 0.50) WC ∆: −1.7 (−4.7, 1.4), ES = 0.15 (*p* = 0.27)		SSB: −0.5 (−2.5, 1.6), ES = 0.12 (*p* = 0.65) Fresh fruit: 3.0 (−1.5, 7.6), ES = 0.33 (*p* = 0.18) Moderate PA: 3.8 (−34.8, 42.2), ES = 0.08 (*p* = 0.84) SSR wknd: −0.7 (−5.8, 4.4), ES = 0.08 (*p* = 0.78) SSR wkday: −1.1 (−5.1, 2.8), ES = 0.19 (*p* = 0.56)		3	NR
Rodearmel et al. 2007 [55] USA	BMI z score ∆: −0.027 (−0.075 to 0.022) (*p* = 0.282) WC ∆: 0.463 (& 1.704 to 0.778) (*p* = 0.462) BMI z score I: 67% C: 53% (*p* < 0.05) I: 47% C: 33% (*p* < 0.05)		Steps/day: (*p* < 0.05)		15.6	NR
**Single Component Dietary Interventions**						
Amaro et al. 2006 [32] Italy	I: 0.345 (0.299–0.390) C: 0.405 (0.345–0.465) (*p* = NR)		V: 21.2 (*p* = 0.01), I: 3.7 (3.5–4.1), C: 2.8 (2.4–3.3) PA: I: 2.1 95% C.I (1.9–2.3), C: 2.2 (2.0–2.4)		17.2	NR
Mihas et al. 2009 [49] Greece	BMI: I: 23.3 (2.8) C:24.0 (3.1) (*p* < 0.001)		Energy: I:8112.4 (1412.4), C:8503.3 (1419.3) (*p* < 0.001) Fat: I: 31.3 (4.4), C:35.4 (4.7) (*p* < 0.001) Saturated Fat: I: 8.2 (1.7) (*p* < 0.001), 12.4 (2.4) (*p* < 0.001) F intake: I: 5.9 (4.3) *p* = 0.036		(4.6, 12.4)	NR
Rabiei et al. 2017 [54] Iran	BMI I: 26.82 (1.42), C: 27.19 (1.55) (*p* = 0.17), I: 26.7 (1.38), 27.13 (1.56) (*p* = 0.09) FU		*n*/A		NR	NR
Viggiano et al. 2015 [67] Italy	Significant in Middle School at 6 mo (*p* = 0.007). Significant in high schools at 6 mo (*p* < 0.001) and at 18 mo (*p* = 0.015).				(30.7, 66.4)	NR
**Single Component Environmental Interventions**						
French et al. 2011 [27] USA	BMI z score ∆: 0.0638 (0.10) (*p* = 0.53)		Significant F&V: 0.4658 (0.23) (*p* = 0.05) TV time: −14.45 (11.79) (*p* = 0.23) Fast food: 0.3847 (0.35) (*p* = 0.27) SSB: −0.0071 (0.16) *p* = 0.96 Snack/sweet: 0.1879 (0.26) *p* = 0.48		3.3	NR
**Single Component Physical Activity Interventions**						
Lindgren et al. 2011 [44] Sweden	NS, ES NR I: 21.9 (14.3–37.2), C: 23.2 (16.1–32) (*p* = 0.696).		PF I: 38.0 (19–86), C: 42 (22–69) (*p* = 0.675).		43.6	NR
Lubans et al. 2011 [45], 2016 [48] Australia	BMI ∆: 0.07 (−34, 38) (*p* = 0.656) BMI z-score ∆: 0.04 (−0.07, 0.14) (*p* = 0.485), WC ∆: 0.3(−0.71, 1.4) (*p* = 0.549)		Screen Time: −32.2 (−53.6, −10.8) (*p* = 0.003) SSB = 0.2 (−0.04, 0.7) (*p* = 0.561) MVPA %: 0.1 (−0.8, 1.0) (*p* = 0.805)		10 (18)	NR
Simons et al. 2014 [58], 2015 [57] Netherlands	BMI-SDS: β = 0.074, 95%CI: 0.008,0.14 Sum of skinfolds: β = 3.22, 95%CI: 0.27,6.17		Non-active video game time: β = −1.76, 95%CI: −3.20,−0.32 Total Sed ST: β = 0.81,95%CI:0.74,0.88 SSB, OR = 0.65 (0.41;1.03), Snacks OR = −1.12 (−2.75,0.50)		10 (4.8)	No effect
Smith et al. 2014 [63,64] Australia	BMI ∆: 0.06 ± 0.12 (*p* = 0.84) WC ∆: 0.5 ± 0.45 (*p* = 0.16)		Screen Time: −30 ± 10.08, (*p* = 0.03) SSB: −0.6 ± 0.26, (*p* = 0.01)		(18.8, 26.3)	NR
Weeks et al. 2012 [22] Australia	BMI: I: 20 (3.5), 20.5 (3.3) C: 20 (3.5), 20.4 (3.7) *p* = 0.895 Weight: I: 53.4 (12.4), 56.6 (12) (*p* = 0.09), C: 53.2 (11.9), 56.5 (13.1) (*p* = 0.951) Lean mass: I: 34,699 (7110), 36,993 (7591), C: 31,993 (4221), 32,974 (5148) (*p* = 0.002)				18.2	NR

BMI, body mass index; C, control; I, Intervention; min, minute; NR, not reported; PA, physical activity; SSB, Sugar Sweetened Beverages; wk, week; yr, years; d, day; hr, hours; mo, months; SB, sedentary behaviour; ∆, change; F, fruit; FF, Fast food; VE, vigorous exercise; F & V, fruit and vegetable; OR, odds ratio; BF, breakfast; JF, junk food; F, fruit; V, Vegetables; Es, effect size; NR: not reported; BSF, biceps skinfold; EB, energy balance; S & D, snacks and desserts.

**Table 3 ijerph-17-06059-t003:** Overall assessment of quality in seven studies (2763 participants in total) in the systematic review of effectiveness of prevention interventions for adolescents from ethnic/racial minorities using the Grading of Recommendations Assessment, Development and Evaluation system.

Category	Rating with Reasoning
Limitations	−2 quality due to limitations
Consistency	No subtraction
Directness	−1 quality level due to population
Precision	−1 due to lack of precision
Publication	−1 quality levels, as publication bias cannot be ruled out
Overall Quality	Low: effect confidence is limited

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
