# Peer review of "Effectiveness of Lifestyle Interventions for Prevention of Harmful Weight Gain among Adolescents from Ethnic Minorities: A Systematic Review"

_ijerph, 2020, doi:10.3390/ijerph17176059_

Round 1

Reviewer 1 Report

this is an interesting paper.

it is difficult to read especially for tables , it is my opinion that you can delete some tables.

the field is important especially in this period where obesity is one of the most important disease is pediatric age

i have a criticisms: i think that actual medicine and research should be focused better about gender differences. 

in obesity, especially for surgical results, gender is essential for primary and secondary end point. So, is it possible to have some sentences into the discussion about this filed? the influence of gender on obesity?

Author Response

Dear Reviewer 1,

Thank you for your comments. Please see our point-by-point response below:

this is an interesting paper.

Thank you.

it is difficult to read especially for tables, it is my opinion that you can delete some tables.

We have removed Table 3A and Table 3B from the main text and have made them supplementary tables instead.

the field is important especially in this period where obesity is one of the most important disease is pediatric age

Thank you.

i have a criticisms: i think that actual medicine and research should be focused better about gender differences. 

in obesity, especially for surgical results, gender is essential for primary and secondary end point. So, is it possible to have some sentences into the discussion about this filed? the influence of gender on obesity?

We have added any information about gender differences in lines 267 to 270: “All six studies that were successful in establishing a difference in BMI/BMI z-score were carried out for both genders. Of the six, four conducted subgroup analyses by gender; two showed no effect; one resulted in more weight loss in boys and the other in girls.”

Reviewer 2 Report

The manuscript entitled “Effectiveness of lifestyle interventions….a systemic review” describes the lifestyle interventions for weight gain between adolescents from different ethnic minorities. The adolescent period provides a lifestyle which impacts in adulthood. The authors have used seven electronic databases to collect the data from 2005 to 2019. The results discuss the seven studies, which showed the weight changes, and none of the studies did not show any overall BMI z-score. There are few significant concerns about the study are 

  1. The authors mentioned the omission of the year before 2005 based on technology, lifestyle, diet, and physical activity. On what basis the exclusion criteria defined lifestyle, diet, and physical activity.
  2. The primary concern is that two essential databases (PubMed and Scopus) were not used in the study, which omits more studies from the analysis. These two database covers more number of articles when compared to the seven of the database. 
  3. In the results section, 7,205 items excluded from this study, please write the exclusion criteria specific to this. If all exclusion criteria noted in the methods used, then specify it in the result section. 
  4. After the second screening, the 45 articles 15 were excluded from the study. Specify in the results section the exclusion criteria used.

Author Response

Dear Reviewer 2,

Thank you for your comments. We have provided a point-by-point response below:

describes the lifestyle interventions for weight gain between adolescents from different ethnic minorities. The adolescent period provides a lifestyle which impacts in adulthood. The authors have used seven electronic databases to collect the data from 2005 to 2019. The results discuss the seven studies, which showed the weight changes, and none of the studies did not show any overall BMI z-score. There are few significant concerns about the study are 

 The authors mentioned the omission of the year before 2005 based on technology, lifestyle, diet, and physical activity. On what basis the exclusion criteria defined lifestyle, diet, and physical activity.

Perhaps the reviewer did not understand that a previous review had been conducted until February 2005 so we did not want to duplicate any results. A more contemporary review will pick up changes in food environments and technology e,g. Smartphones, apps and physical activity trackers that are relevant for policy makers and practice now.

Please see lines 104-108: “A comprehensive review of weight gain prevention interventions in adolescents was published in 2005 [19]. We selected 2005 as our start date because of both the previous review and acknowledging the significant advancements in technology and health promotion that have occurred since then with nearly 100% of public schools in the US having access to Internet in 2005 [20]”.

  1. The primary concern is that two essential databases (PubMed and Scopus) were not used in the study, which omits more studies from the analysis. These two database covers more number of articles when compared to the seven of the database. 

The search strategy was overseen by a very experienced librarian who has advised on hundreds of reviews in nutrition and medicine.

Please note that Pubmed is not a database but a search engine for Medline library so those articles are all captured.

Scopus is a commercial database and it can be at their discretion as to what articles they include (preference for Elsevier) and exclude. We grant it is very extensive. You will see that generally Cochrane reviews on topics related to this have not used Scopus but like us have consulted a large number of databases to capture articles that are linked with government and non-commercial sources. We have also stated that technology-mediated studies were of interest and we know that Scopus excluded the second highest ranking technology and health journal for many years.

  1. In the results section, 7,205 items excluded from this study, please write the exclusion criteria specific to this. If all exclusion criteria noted in the methods used, then specify it in the result section. 

The 7,205 items were excluded after initial screening based on title and abstract as they failed to meet the inclusion criteria.

Please see lines 149-151: “Duplications were removed, leaving 7,409 abstracts for the first screening, thereafter another 7,205 were excluded as they failed to meet the inclusion criteria i.e. not an intervention, not the primary outcome, not the study design.”

  1. After the second screening, the 45 articles 15 were excluded from the study. Specify in the results section the exclusion criteria used.

Thank you for your comment. Contrary to your statement/understanding, 15 papers were not excluded. All 45 papers were included. Of the 45 papers included, 30 were individual studies included in the review with the remaining 15 papers to be secondary papers (protocol, long term follow-up etc.) as stated in Line 151, page 4.

We have included the list of full-text articles excluded and reasons for exclusion in Table S2. Please see lines 156 -157: Please see Table S2 for the list of full text articles excluded and supported reasoning.

Round 2

Reviewer 2 Report

Answered all the comments.